# Piezoelectricity in hafnia

Sangita Dutta [1,2✉], Pratyush Buragohain [3], Sebastjan Glinsek [1], Claudia Richter[4], Hugo Aramberri [1], Haidong Lu[3], Uwe Schroeder [4], Emmanuel Defay[1], Alexei Gruverman [3✉] & Jorge Íñiguez [1,2✉]

Because of its compatibility with semiconductor-based technologies, hafnia (HfO$_2$) is today's most promising ferroelectric material for applications in electronics. Yet, knowledge on the ferroic and electromechanical response properties of this all-important compound is still lacking. Interestingly, HfO$_2$ has recently been predicted to display a negative longitudinal piezoelectric effect, which sets it apart from classic ferroelectrics (e.g., perovskite oxides like PbTiO$_3$) and is reminiscent of the behavior of some organic compounds. The present work corroborates this behavior, by first-principles calculations and an experimental investigation of HfO$_2$ thin films using piezoresponse force microscopy. Further, the simulations show how the chemical coordination of the active oxygen atoms is responsible for the negative longitudinal piezoelectric effect. Building on these insights, it is predicted that, by controlling the environment of such active oxygens (e.g., by means of an epitaxial strain), it is possible to change the sign of the piezoelectric response of the material.

[1] Materials Research and Technology Department, Luxembourg Institute of Science and Technology, 5 avenue des Hauts-Fourneaux, L-4362 Esch/Alzette, Luxembourg. [2] Department of Physics and Materials Science, University of Luxembourg, 41 Rue du Brill, Belvaux L-4422, Luxembourg. [3] Department of Physics and Astronomy, University of Nebraska-Lincoln, Lincoln, NE 68588-0299, USA. [4] NaMLab gGmbH, Noethnitzer Strasse 64 a, 01187 Dresden, Germany. ✉email: sangita.dutta@list.lu; agruverman2@unl.edu; jorge.iniguez@list.lu

Hafnia (HfO$_2$) is a well-known material in the electronics industry, since its introduction in 2007 by Intel as a convenient gate dielectric for field-effect transistors (FETs)[1]. The announcement of ferroelectricity in this compound a few years later[2,3] caused great excitement, as it opened the door to the development of (inexpensive, easy to process) electronic devices that could benefit from a switchable polarization, e.g., memories based on ferroelectric FETs. Ever since, a lot of efforts have focused on understanding and controlling ferroelectricity in HfO$_2$, taking advantage of the unique possibilities it may offer (e.g., ferroelectric negative-capacitance effects[4]) and advancing towards commercial devices. By now, ferroelectric hafnia has gathered the interest of engineers, materials scientists, and physicists alike, being one of today's best studied and most promising materials.

As compared to traditional soft-mode ferroelectrics (e.g., perovskite oxides like PbTiO$_3$ or BaTiO$_3$[5]), hafnia displays many peculiar features that we are only starting to understand. For example, theoretical work suggests that hafnia's ferroelectricity may not be proper in character[6–8], yet it is switchable, which sets this compound apart from all ferroelectrics used so far in applications. Further, the nature of its (anti)polar instabilities is such that very narrow domains, and very narrow domain walls, occur naturally in it[9,10]; in effect, this makes HfO$_2$ a quasi-2D ferroelectric, and may explain the resilience of its polar phase in nanometric samples. In fact, unlike traditional materials, HfO$_2$ seems to improve its ferroelectric properties as the samples decrease in size; moreover, the first reports of ferroelectricity in thick films or bulk samples are very recent[11,12]. In sum, from both applied and fundamental perspectives, ferroelectric hafnia is revealing itself as a very interesting compound.

Comparatively, the electromechanical response properties of ferroelectric HfO$_2$ have received little attention so far, although we believe this situation will quickly change. Indeed, the processing advantages that hafnia offers (as compared to perovskite oxides) make it a viable candidate for applications as a piezoelectric (e.g., in piezotronics, radio-frequency filters) where it might potentially compete with wurtzite compounds (e.g., AlN, ZnO).

At a more fundamental level, it has been recently predicted from first principles[13,14] that the usual ferroelectric phase of HfO$_2$ (orthorhombic with space group $Pca2_1$) presents a negative longitudinal piezoresponse, i.e., that compressing the material along the polarization direction will result in an enhancement of its polar distortion. If confirmed, this property would widen even more the gap between HfO$_2$ and the ferroelectric perovskite oxides, all of which behave in exactly the opposite way[15]. Intriguingly, though, existing experimental measurements of hafnia's piezoresponse[11] suggest a perovskite-like behavior (i.e., a positive longitudinal effect), and thus contradict the first-principles predictions. Further, we still lack a satisfying physical picture explaining the atomistic origin of the predicted negative piezoresponse, a simple understanding that would allow us to propose ways to control and optimize the effect. Hence, in our opinion, the piezoelectric response of hafnia is an open problem.

Here we present a first-principles and experimental investigation of the piezoelectric properties of HfO$_2$. First, we confirm the negative longitudinal effect, from both theory and experiment, notwithstanding the fact that the experimental result is sample dependent. Second, based on our first-principles simulations, we provide a simple and plausible explanation of the atomistic mechanisms controlling the effect. Further, based on this understanding, we predict that the ferroelectric phase of hafnia can be modified (by epitaxial strain) to either enhance or reduce the negative longitudinal piezoresponse, or even change its sign. We conclude with a brief discussion of the implications of our results, and an outlook of the challenges and opportunities ahead.

## Results and discussion

In the following we present our simulation and experimental results. In most occasions we discuss in parallel our findings for HfO$_2$ and the corresponding results for a representative ferroelectric perovskite, which allows us to better highlight the specificity of hafnia as compared with classic materials. For the perovskite, we consider PbTiO$_3$ at the theoretical level and PbZr$_{1-x}$Ti$_x$O$_3$ (PZT, with $x = 0.6$) at the experimental level. In our experimental presentation, we also show results for polyvinylidene fluoride (PVDF), a compound with a well-characterized negative longitudinal piezoresponse[16,17].

**Basic first-principles predictions**. We start our first-principles investigation by relaxing the most usual ferroelectric polymorph of HfO$_2$, with $Pca2_1$ space group. The obtained solution (Supplementary Table 1) agrees well with previous results in the literature[18,19]. The ferroelectric polarization characterizing the $Pca2_1$ phase can be appreciated by comparing the cubic paraelectric structure of Fig. 1a with the polar state in Fig. 1b: it is caused by the downward shift of the O$_I$ anion sublattice, which results in a positive polarization along the third Cartesian direction ($c$ axis in the figure). Note that all Hf cations in the

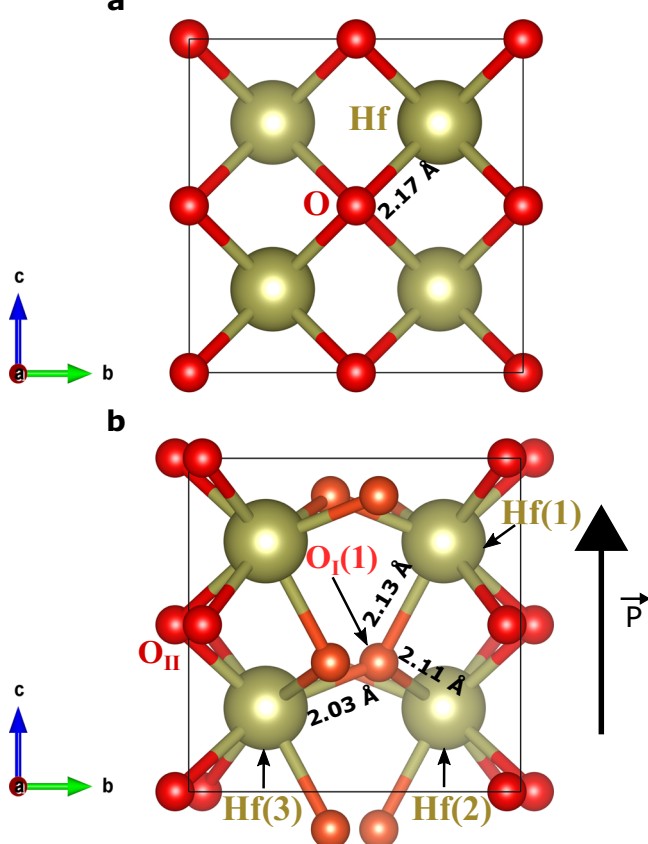

**Fig. 1 Key structural features of HfO$_2$.** Structure of the cubic paraelectric (**a**) and orthorhombic ferroelectric (**b**) polymorphs of HfO$_2$. In the cubic $Fm\bar{3}m$ phase, all Hf and O atoms are equivalent by symmetry. In the ferroelectric $Pca2_1$ structure we have two symmetry-inequivalent sets of oxygen atoms –labeled O$_I$ (shown in orange) and O$_{II}$ (red), respectively–, while all Hf atoms are equivalent. In panel **b**, the black arrow indicates the positive spontaneous polarization of the structure shown, which is essentially related to the vertical downward shift of the O$_I$ atoms from their high-symmetry position in the cubic phase. In the ferroelectric state, the O$_I$ atoms have three nearest-neighboring Hf cations; the bonding distances are explicitly indicated for the O$_I$(1) atom, which is a representative case.

**Table 1 Calculated piezoelectric tensors for HfO$_2$.**

|  | Index | $\bar{e}$ | $e$ | $d$ |
|---|---|---|---|---|
| VASP | 31 | −0.37 | −1.31 | −1.71 |
|  | 32 | −0.34 | −1.33 | −1.77 |
|  | 33 | 0.62 | −1.44 | −2.51 |
|  | 15 | −0.28 | −0.20 | −2.03 |
|  | 24 | −0.20 | 0.64 | 6.74 |
| ABINIT | 31 | −0.39 | −1.53 | −2.71 |
|  | 32 | −0.36 | −1.40 | −1.66 |
|  | 33 | 0.65 | −1.34 | −1.64 |
|  | 15 | −0.29 | −0.23 | −2.70 |
|  | 24 | −0.22 | 0.69 | 6.60 |

We show the total (**e**) and frozen-ion ($\bar{e}$) direct piezoelectric tensor (in C m$^{-2}$), as well as the total converse piezoelectric tensor **d** (in pm V$^{-1}$). Indices given in Voigt notation.

**Table 2 Same as Table 1, but for the ferroelectric phase of PbTiO$_3$.**

| Index | $\bar{e}$ | $e$ | $d$ |
|---|---|---|---|
| 31 | 0.23 | 1.62 | −39 |
| 33 | −0.30 | 4.95 | 208 |
| 15 | 0.03 | 4.58 | 78 |

polar phase are equivalent by symmetry and occupy a Wyckoff position that we label 'Hf'. In contrast, the polar state features two symmetry-inequivalent oxygen sublattices, which we label 'O$_I$' and 'O$_{II}$', respectively. Also, in Fig. 1b and the following we use a number in parenthesis (e.g., 'Hf(2)' or 'O$_I$(1)') to label individual atoms belonging to a particular sublattice.

To fix ideas, noting that the sign of the piezoelectric coefficients depends on the sign of the spontaneous polarization[20], in the following we always work with the ferroelectric state with positive polarization as shown in Fig. 1b. We calculate this polarization to be $P_3 = 54.75$ μC cm$^{-2}$, in agreement with previous literature[18,19].

We now consider the piezoelectric tensor[21]

$$e_{\alpha j} = \frac{\partial P_\alpha}{\partial \eta_j}, \qquad (1)$$

where $P_\alpha$ is the polarization component along Cartesian direction $\alpha$ and $\eta_j$ is a symmetric strain labeled using Voigt notation[21]. For analysis purposes, it is convenient to decompose the piezoelectric response into a frozen-ion contribution ($\bar{e}_{\alpha j}$) and a lattice-mediated part (defined as the difference $e_{\alpha j} - \bar{e}_{\alpha j}$). $\bar{e}_{\alpha j}$ is obtained by freezing the atoms in their unperturbed equilibrium positions, so they cannot respond to the applied strain; it thus captures a purely electronic effect. Finally, from knowledge of the $e_{\alpha j}$ and the elastic constants $C_{jk}$, one can obtain

$$d_{\alpha j} = \frac{\partial P_\alpha}{\partial \sigma_j} = (C^{-1})_{jk} e_{\alpha k} = S_{jk} e_{\alpha k} \qquad (2)$$

where $\sigma_j$ is the $j$-th component of an applied external stress (in Voigt notation), $\mathbf{S} = \mathbf{C}^{-1}$ is the elastic compliance, and we assume summation over repeated indices. Note that this piezo-response **d** tensor is of interest, as it is the one most easily accessible in experiment and the one usually exploited in applications. Computing the piezoelectric tensors is straightforward using density functional perturbation theory (DFPT)[22].

Table 1 shows the results we obtain for **e**, $\bar{e}$, and **d** using two different—but essentially equivalent, both accurate—implementations of density functional theory (DFT). (We attribute the existing numerical differences mainly to the use of different pseudopotentials; see the Experimental Section.) We confirm a negative value of the $e_{33}$ coefficient, indicating that a positive strain (stretching) of the unit cell along the polar direction will yield a reduction of the polarization. (Recall that we work with an unperturbed state with $P_3 > 0$ and $P_1 = P_2 = 0$.) We further verify this result by performing a finite-difference calculation of the change in $P_3$ upon application of a small strain $\eta_3$. Our results are also in agreement with the DFT predictions previously published[14].

It is interesting to note that the lattice-mediated response is always larger than the frozen-ion response. In particular, Table 1 clearly shows that the lattice response is responsible for the negative value of $e_{33}$.

Supplementary Table 2 shows the results obtained for the elastic and compliance tensors, which allow us to compute the **d** tensor in Table 1. While the relationship between **e** and **d** is not trivial in materials with a relatively low symmetry (as is the case of ferroelectric HfO$_2$), we do obtain a negative $d_{33} = -2.51$ pm V$^{-1}$. Hence, we predict that, upon application of a compressive stress $\sigma_3 < 0$, $P_3$ will increase.

Table 2 shows results for the piezoelectric tensors of the ferroelectric phase of PbTiO$_3$ (tetragonal, with space group $P4mm$). We find that the $e_{\alpha j}$ coefficients are generally larger for the perovskite than for HfO$_2$ (by a factor of up to 4, if we focus on their absolute values). Interestingly, the difference becomes much greater for the $d_{\alpha j}$ coefficients, as PbTiO$_3$ presents values between 1 and 2 orders of magnitude larger than those of hafnia: for example, we get a $d_{33}$ of 208 pm V$^{-1}$ for PbTiO$_3$ and −2.51 pm V$^{-1}$ for HfO$_2$. The obtained giant $d_{33}$ response of lead titanate agrees with previous experimental[23] and theoretical[24] reports. Interestingly, our calculations indicate that the main reason behind this result is the softness of the compound along the polarization direction: we get $C_{33} = 51.8$ GPa and $S_{33} = 48.72$ TPa$^{-1}$ for PbTiO$_3$ (the full tensors are given in Supplementary Table 3), which contrast with the much stiffer case of HfO$_2$ ($S_{33} = 2.97$ TPa$^{-1}$, see Supplementary Table 2).

When considering the experimental manifestation of the negative $d_{33}$ predicted for hafnia, one has to take into account an important feature of most samples: they are polycrystalline. Hence, typically, the measured piezoelectric response will not correspond to a unique well-defined orientation, but to an effective average. If we assume a sample composed of randomly oriented grains, and poled so that all grains display a polarization with a positive $P_3$ component, we can estimate an effective $d_{33,\mathrm{eff}}$ as[25,26]

$$
\begin{aligned}
d_{33,\mathrm{eff}} &= \left\langle \cos\theta \left[ (d_{15}+d_{31})(\sin^2\theta\sin^2\varphi) + (d_{24}+d_{32})(\sin^2\theta\cos^2\varphi) + d_{33}\cos^2\theta \right] \right\rangle \\
&= (d_{15}+d_{31})\langle\cos\theta\sin^2\theta\rangle\langle\sin^2\varphi\rangle + (d_{24}+d_{32})\langle\cos\theta\sin^2\theta\rangle\langle\cos^2\varphi\rangle + d_{33}\langle\cos^3\theta\rangle \\
&= \frac{1}{3\pi}(d_{15}+d_{31}+d_{24}+d_{32}) + \frac{4}{3\pi}d_{33},
\end{aligned}
$$
(3)

where $\langle\ldots\rangle$ indicates an average over the Euler angles $\varphi$ and $\theta$, which span all possible orientations with $P_3 > 0$ (i.e., $0 < \theta < \pi/2$ and $0 < \varphi < 2\pi$). By calculating this average we obtain $d_{33,\mathrm{eff}} = -0.94$ pm V$^{-1}$, suggesting that even in polycrystalline HfO$_2$ samples we expect to measure a negative longitudinal piezo-response. Hence, to summarize: according to our DFT calculations, the $d_{33}$ response of HfO$_2$ is predicted to be somewhere between −2.5 pm V$^{-1}$ (single-crystal limit) and −0.9 pm V$^{-1}$ (untextured fully poled polycrystalline limit).

**Experimental confirmation.** To determine experimentally the sign of the effective piezoelectric coefficient $d_{33,\mathrm{eff}}$ in hafnia, we carry out comparative dynamic piezoelectric measurements by means of piezoresponse force microscopy (PFM), using materials with known piezoelectric coefficients as a reference. In PFM,

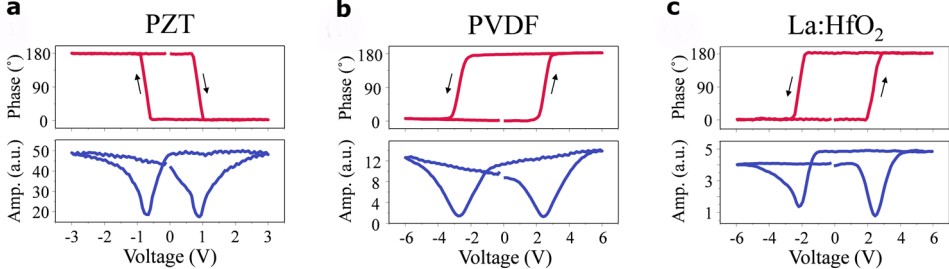

**Fig. 2 Dynamic measurements of piezoelectricity using piezoresponse force microscopy (PFM). a–c** PFM phase (top panel) and amplitude (bottom panel) loops measured in the IrO$_2$/PZT/Pt capacitor (**a**), PVDF film (**b**), and Pt/Ti/TiN/La:HfO$_2$/TiN capacitor (**c**). The loops were obtained in the bias-off mode to minimize the electrostatic contribution to the PFM signal.

application of an alternating (ac) electric field to the sample via a conductive tip results in an oscillation with the frequency of the applied field, due to the converse piezoelectric effect[27,28]. The amplitude and phase of the oscillation provide information about the magnitude and sign of $d_{33,eff}$, respectively (see Supplementary Note 1 and Supplementary Figs. 1–3).

In a material with a positive longitudinal piezoresponse, such as PZT[29], the sample oscillation will be in phase with the driving electric field when the polarization is oriented downward, while it will be in anti-phase when the polarization is oriented upward. In addition, in the switching spectroscopy mode of PFM, a pulsed low-frequency triangular waveform is superimposed on the ac waveform to generate a piezoelectric strain hysteresis loop related to the local polarization reversal[30]. Thus, the sense of rotation of the PFM phase loops in Fig. 2 is directly related to the sign of the $d_{33,eff}$. For example, a clockwise rotation of the PFM phase signal is indicative of a positive $d_{33,eff}$ coefficient, as is illustrated by the PFM phase hysteresis loop measured in the IrO$_2$/PZT/Pt capacitor in Fig. 2a. In this case, the phase signal is in phase (anti-phase) with the ac modulation field at the far positive (negative) dc bias, which generates the downward (upward) orientation of the polarization. In contrast, in a material with a negative $d_{33,eff}$, such as PVDF[17,31], the phase signal is in phase (anti-phase) with the ac modulation field at the far negative (positive) dc bias corresponding to the upward (downward) orientation of the polarization, resulting in anti-clockwise rotation in the PFM phase loop (Fig. 2b). (See the Experimental Section for details on our PZT and PVDF samples). Under the same conditions, as shown in Fig. 2c, the PFM phase loop measured in TiN/La:HfO$_2$/TiN capacitors (with 20 nm-thick La:HfO$_2$) exhibits an anti-clockwise rotation similar to that of PVDF films and opposite to that of PZT capacitors. (See the Experimental Section for details on our HfO$_2$ samples; note that similar results were obtained for 10 nm-thick samples.) Clearly, this behavior is indicative of the negative sign of $d_{33,eff}$ in the La:HfO$_2$ film. We estimate $d_{33,eff}$ to be between $-2$ pm V$^{-1}$ and $-5$ pm V$^{-1}$ (Supplementary Fig. 2), in excellent agreement with our theoretical result between $-0.9$ pm V$^{-1}$ (in the polycrystalline disordered limit) and $-2.5$ pm V$^{-1}$ (in the single-crystal limit).

It is important to note here that evidence for a positive longitudinal piezoresponse $d_{33,eff}$ in hafnia has been reported in the experimental literature, for example, for ultrathin (10 nm) Si-doped HfO$_2$ films[32], for thicker (70 nm) Y-doped films[33], or for La-doped films with thicknesses up to 1 μm[11]. Indeed, in the course of this work, we found ourselves that the application of the same experimental protocol to other HfO$_2$ films (thicker, grown by different means) yields a positive longitudinal effect. Hence, one may wonder what is especial about the La:HfO$_2$ films for which we get a negative effect. May it be extrinsic?

Interestingly, the La:HfO$_2$ films studied here were characterized in ref. [34] using grazing-incidence X-ray diffraction. In that work,

it was determined that these La:HfO$_2$ films present a strong out-of-plane texture, leading to relatively high remnant polarization compared to other dopants. This stronger texture may cause local strains in the films, and potentially act as an extrinsic factor affecting the local electromechanical properties. However, since all the domains measured in our study showed a negative response, we tend to believe that the effect observed in this work is intrinsic.

As far as we can tell, there is only one other publication suggesting a similar negative longitudinal piezoresponse. In 2019 Chouprik et al.[35] reported 'anomalous' switching in 10 nm-thick films of Hf$_{0.5}$Zr$_{0.5}$O$_2$, which in PFM appeared as a polarization reversal against the applied electric field, and could thus be interpreted as a negative piezoresponse. This behavior was observed in about 20% of the domains of the pristine films.

Hence, there is experimental evidence that the piezoelectric properties of HfO$_2$ are sample and sample-history dependent. This suggests that a careful and systematic characterization will be needed to determine the factors (intrinsic or extrinsic) controlling the piezoresponse, including its sign.

**Origin of the negative piezoresponse.** Our DFPT calculations allow us to track down the computed negative value of $e_{33}$. As presented in Supplementary Note 2, within perturbation theory[22] we write the piezoelectric tensor as

$$e_{\alpha j} = \bar{e}_{\alpha j} + \Omega_0^{-1} Z_{m\alpha} (\Phi^{-1})_{mn} \Lambda_{nj} , \qquad (4)$$

where the second term on the right-hand side of this equation shows that the lattice-mediated part of **e** depends on the unit-cell volume of the unperturbed system ($\Omega_0$), the Born effective charge tensor (**Z**, which quantifies the polarization change caused by atomic displacements), the force-constant matrix (**Φ**, i.e., the second derivatives of the energy with respect to atomic displacements) and the force-response internal strain tensor (**Λ**, which quantifies the atomic forces that appear when a strain is applied). Here, $m$ is a combined index that runs over all atoms in the unit cell and the three spatial directions.

By inspecting the calculated tensors for HfO$_2$, and by comparing with those obtained for PbTiO$_3$, we can identify the atomistic underpinnings of the sign of $e_{33}$.

First off, let us note that there is nothing peculiar concerning the force-constant matrices **Φ**: for both materials, these matrices reflect the fact that the ferroelectric phase is a stable equilibrium state. Hence, they are positively defined tensors without any feature that is relevant to the present discussion.

As for the Born effective charges **Z** (Supplementary Table 4), they have the expected signs and are relatively large in magnitude: we get values over $+5$ for Hf and below $-2.5$ for O, exceeding the nominal respective charges of $+4$ and $-2$. This feature reflects a mixed ionic-covalent character of the chemical bonds in the material, and is typical of other ferroelectrics like, e.g., PbTiO$_3$

**Table 3 Λ tensors for the symmetry-inequivalent atoms of the ferroelectric phase of HfO₂ (in eV Å⁻¹).**

| | | | | | | |
|---|---|---|---|---|---|---|
| Hf | −4.02 | 0.00 | −2.29 | 8.39 | 0.83 | 0.79 |
| | −1.04 | 11.52 | −1.40 | 4.50 | 3.58 | 5.87 |
| | −1.87 | 1.45 | **−4.07** | −2.83 | −2.61 | 3.42 |
| O$_I$ | 2.06 | −0.63 | 1.09 | −5.87 | −3.29 | 0.91 |
| | −1.97 | −1.75 | −1.43 | −2.77 | −4.60 | −2.34 |
| | −1.61 | 0.42 | **3.22** | 1.36 | 0.29 | −4.53 |
| O$_{II}$ | −5.87 | 1.10 | −1.07 | 4.67 | 2.46 | 0.06 |
| | 0.79 | 7.87 | 0.54 | −1.71 | 4.10 | 0.52 |
| | 3.46 | 1.86 | **0.85** | 1.27 | −1.36 | 5.44 |

The 3 rows correspond, respectively, to the 3 spatial directions; the 6 columns correspond, respectively, to the 6 strain indices in Voigt notation. Marked in bold are the entries controlling the longitudinal piezoresponse, as discussed in the text.

itself (see the Born charges we obtain for PbTiO₃ in Supplementary Table 5). In addition, because of the relatively low site symmetries in the ferroelectric phase of HfO₂, in this compound, the charge tensors present small non-zero off-diagonal components. While this feature does set HfO₂ apart from PbTiO₃, we checked it has no influence in the sign of $e_{33}$. In conclusion, the **Z** tensors do not explain the differentiated behavior of $e_{33}$ in these two compounds.

Hence, we are left with the **Λ** tensors, which are given in Table 3 and Supplementary Table 6 for HfO₂ and PbTiO₃, respectively. Let us focus on the "33" entries for each of the atom-specific tensors, i.e., the numbers quantifying the atomic force along direction 3 (parallel to the polarization) caused by a positive strain $\eta_3 > 0$ (stretching). In the case of PbTiO₃, the strain-induced forces are positive for the cations (Pb and Ti) and negative for the two symmetry-inequivalent oxygens in the unit cell. This means that, in response to the vertical stretching of the cell, the cations will tend to move up while the oxygens will tend to move down. Since the unperturbed state has $P_3 > 0$, this movement will clearly yield an increase of the polarization; hence, we have $e_{33} > 0$.

The situation is just opposite for HfO₂: in this case, a stretching of the cell ($\eta_3 > 0$) causes the Hf cations to move down and the oxygens (particularly those of type I) to move up. Since the starting point has $P_3 > 0$, and since the Born charges in HfO₂ have the natural signs for cations and anions, these strain-induced displacements will yield a reduction of the magnitude of the polarization. This is indeed reflected in our computed $e_{33} < 0$; we have thus identified the atomistic origin of the effect.

**Physical insight**. While the above discussion is clear from a numerical point of view, it hardly provides us with a satisfying physical understanding. Can we rationalize the mechanisms controlling the sign of the strain-induced forces and, thus, of $e_{33}$?

In perovskite oxides, it is known that the structural instabilities of the parent cubic phase (as, e.g., those leading to ferroelectricity) are largely determined by steric and ion-size aspects, usually discussed in terms of simple descriptors such as the Goldschmidt tolerance factor[36]. Ultimately, these effects are a reflection of the ions' tendency to optimize the chemical bonds in their first (nearest neighbor) coordination shell, as successfully captured by phenomenological theories such as, for example, the bond-valence model[37].

These bonding considerations readily allow us to understand the piezoelectric response in PbTiO₃, a simple model case. In this compound, the cubic paraelectric phase presents Ti and Pb cations that are equidistant to 6 and 12 first-neighboring oxygens, respectively. Then, as shown in Fig. 3 (for a state with $P_3 > 0$), the

ferroelectric distortion results in a tetragonal structure where the cations reduce the number of closest oxygens neighbors. For the sake of simplicity, let us focus on the case of the central Ti cation, which passes from being 6-fold coordinated in the paraelectric phase (panel a) to having only 5 close oxygens in the ferroelectric state (panel b). In fact, among these oxygens, there is one (the apical type I oxygen that lies above Ti) forming the shortest (and strongest) Ti–O bond. (We know the details of this bond from previous theoretical works on PbTiO₃, which also show that the type II oxygens are mainly bonded to the Pb cations[38].) Imagine we now stretch the cell along the polarization direction ($\eta_3 > 0$), and assume that the atoms will rearrange in order to maintain the preferred length of the strongest bonds. For that to happen, as sketched in Fig. 3c, the central Ti should move up and the mentioned O$_I$ oxygen that bonds with it should move down, which results in an increased polarization. This expectation is in perfect correspondence with our computed **Λ** tensor (see Supplementary Table 4). A similar argument applies to the displacements of the Pb and O$_{II}$ ions in reaction to $\eta_3 > 0$. Thus, this simple picture explains the positive $e_{33}$ obtained for PbTiO₃.

The situation in HfO₂ is harder to analyze, for two main reasons. First, the atomic chemical environments are far more complex than in PbTiO₃ and identifying dominant bonds is not trivial. (Our attempts at a clear-cut quantification—e.g., by inspecting the magnitude of the interatomic force constants— were not convincing enough.) Second, as shown from first principles[6–8], the nature of ferroelectricity in HfO₂ is far more complicated than in PbTiO₃. Nevertheless, an appealing physical picture emerges from our results, as follows.

Let us start by inspecting the distortions connecting the cubic paraelectric phase (a convenient reference for the sake of this argument) and the $Pca2_1$ ferroelectric state of HfO₂. In the cubic phase, all O anions have four nearest-neighboring Hf cations, and all Hf–O bond distances are equal to 2.17 Å. In contrast, in the ferroelectric phase we have two sublattices of symmetry-equivalent oxygens (labeled O$_I$ and O$_{II}$), and all oxygens form relatively short bonds with only three Hf atoms. Let us focus on the O$_I$ sublattice, responsible for the spontaneous polarization. As shown in Fig. 1b for the representative case of the O$_I$(1) atom, we have the following shortest bonding distances: 2.13 Å for Hf(1)–O$_I$(1), 2.11 Å for Hf(2)–O$_I$(1), and 2.03 Å for Hf(3)–O$_I$(1). Further, the computed equilibrium charge density in Fig. 4a, b suggests that O$_I$(1) forms similarly strong bonds with its three neighboring Hf cations.

It is thus apparent that the formation of a reduced number of relatively short Hf–O bonds is the driving force for the stabilization of the ferroelectric state. However, interestingly, the atomic rearrangements yielding the preferred Hf–O coordination do not necessarily contribute to the spontaneous polarization. For example, as indicated in Fig. 3d, e, to reach the optimal configuration the O$_I$ anions shift both vertically (parallel to the polar axis) and horizontally (perpendicular to it). The former displacements yield the spontaneous polarization of HfO₂, while the latter (which follow an anti-polar pattern of sorts) have no contribution to it. As for the O$_{II}$ anions, Fig. 3e shows that they present anti-polar displacements along the horizontal direction, with no contribution to the polarization. These observations suggest that the development of the spontaneous polarization may not be the main driving force for the $Pca2_1$ phase to occur. Indeed, first-principles calculations show that the cubic phase of HfO₂ does not present any ferroelectric instability[6]. Additional first-principles studies suggest that the occurrence of the $Pca2_1$ phase of HfO₂ depends on the prior condensation of a non-polar mode that constitutes a strong instability of the cubic state (and yields a well-known tetragonal polymorph), and that it is further conditioned by a

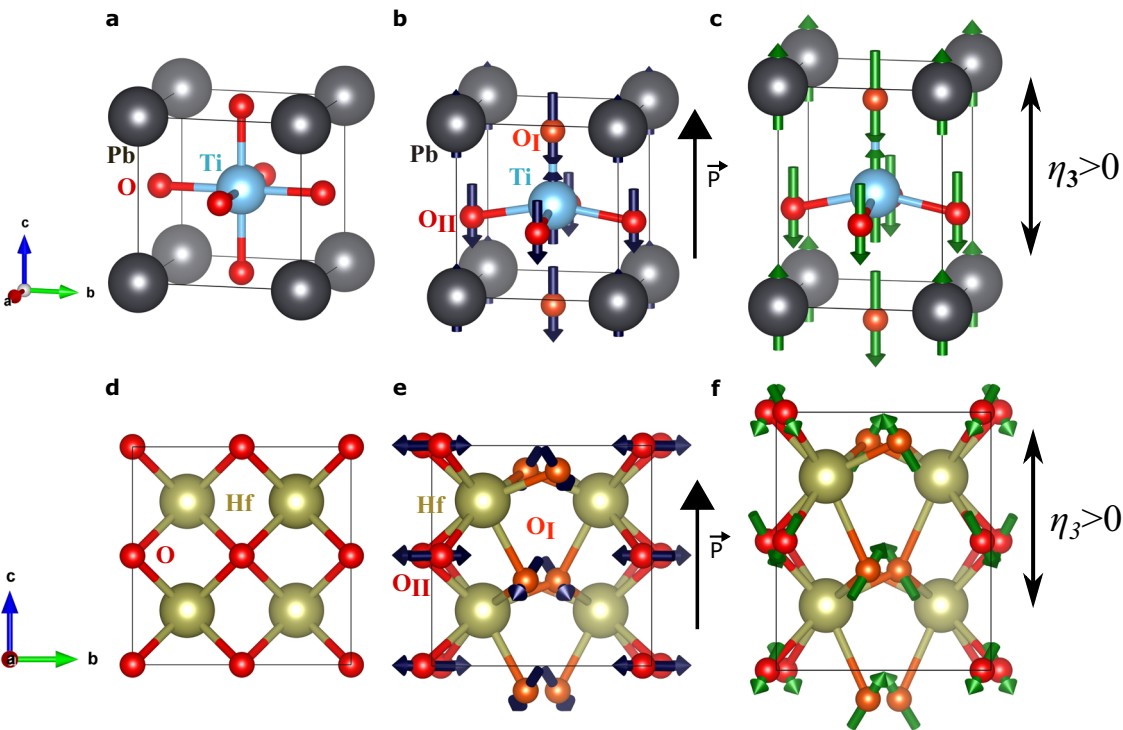

**Fig. 3 Comparison between PbTiO₃ and HfO₂.** Cubic $Pm\bar{3}m$ paraelectric (**a**) and tetragonal $P4mm$ ferroelectric (**b**) phases of PbTiO₃. The tetragonal phase presents two symmetry-inequivalent oxygen anions, colored differently and labeled by $O_I$ and $O_{II}$, respectively. In panel **b** the arrow on the right marks the spontaneous polarization, which is essentially related to the upward displacement of the Pb and Ti cations with respect to the oxygen atoms (the arrows on the atoms mark such displacements). Panel **c** is a sketch of the tetragonal phase subject to a tensile $\eta_3 > 0$ strain (the strain is exaggerated for clarity); the arrows on the atoms indicate how they react in response to the strain, as computed from first principles. Panels **d**–**f** are analogous to the previous three panels, but featuring the paraelectric (**d**) and ferroelectric (**e**) states of HfO₂, and its longitudinal piezoresponse (**f**).

very strong coupling between the polarization and other supplementary (hard) distortions[7,8]. Hence, HfO₂ is qualitatively different from proper ferroelectric perovskites like PbTiO₃.

Then, to understand the $e_{33}$ piezoresponse of HfO₂, we have to discuss how a strain along the polar axis will affect the position of the $O_I$ ions responsible for the spontaneous polarization. By inspection of the atomic environment of the $O_I(1)$ atom (a representative case), we make a critical observation: there is only one Hf–$O_I(1)$ bond clearly aligned with the polar axis, namely, the one connecting $O_I(1)$ with the Hf(1) cation above it (Fig. 1b); this bond will be strongly affected by a vertical $\eta_3$ strain. In contrast, the remaining two Hf–$O_I(1)$ bonds, with Hf(2) and Hf(3), lie largely in the horizontal $ab$ plane, and will be mildly impacted by a vertical strain. Thus, upon application of a tensile strain $\eta_3 > 0$, we can expect $O_I(1)$ to move upward in order to preserve the optimum Hf(1)–$O_I(1)$ distance. This is exactly the behavior we find from first principles, illustrated in Fig. 3f. Interestingly, and somewhat anecdotally, the upward movement of $O_I(1)$ (Fig. 3f) goes against the downward shift of this very same ion when the polar phase condenses (Fig. 3e). Consequently, the polarization is reduced in reaction to $\eta_3 > 0$, which yields a negative $e_{33}$.

The difference between PbTiO₃ and HfO₂ is pictorially illustrated in Fig. 3: in PbTiO₃ the ionic reaction to $\eta_3 > 0$ (panel c) adds to the distortion responsible for the spontaneous polarization (panel d), while in HfO₂ it goes largely against it (for $O_I$, the arrows of panel e are reversed in panel f). Ultimately, it is the peculiar chemical environment of the $O_I$ anions in HfO₂ that is responsible for the effect.

While appealing, this picture may seem speculative. Nevertheless, it suggests that, by controlling the chemical environment of the $O_I$ atoms that dominate the $e_{33}$ response, we may be able to

affect the magnitude of the effect in a very definite way. More precisely: by decreasing the Hf(1)–$O_I(1)$ distance, we may be able to make this bond stronger and, thus, make $e_{33}$ more negative; conversely, by weakening the Hf(1)–$O_I(1)$ link, we should have a response increasingly controlled by the Hf(2)–$O_I(1)$ and Hf(3)–$O_I(1)$ pairs, which should result in a less negative $e_{33}$. We put this hypothesis to a test in the next section.

**Prediction of a tunable piezoresponse.** To control the bonds of interest and monitor their effect on $e_{33}$, we simulate the $Pca2_1$ ferroelectric phase of HfO₂ subject to an isotropic epitaxial strain in the plane perpendicular to the polarization.

We do this by running structural relaxations where the in-plane lattice vectors are constrained to form a 90° angle and their magnitudes fixed to $a = a_0(1 + \eta_{epi})$ and $b = b_0(1 + \eta_{epi})$, where $a_0$ and $b_0$ are the previously obtained equilibrium lattice constants (Supplementary Table 1) and $\eta_{epi}$ the applied epitaxial strain. Our calculations suggest that the $Pca2_1$ orthorhombic phase is an equilibrium energy minimum in a wide $\eta_{epi}$-range, from about −7 % to about +4 %. (We find that beyond this range the ferroelectric polymorph losses its stability and transforms into other structures that are of no interest here.) Figure 5 shows our results for the $\eta_{epi}$ dependence of the Hf–$O_I(1)$ bond lengths, which follow the expected behavior: the Hf(1)–$O_I(1)$ link (which largely lies along the vertical direction) gets longer as we compress in-plane ($\eta_{epi} < 0$), following the growth of the out-of-plane lattice constant $c$ (see Supplementary Fig. 4). Conversely, the Hf(2)–$O_I(1)$ and Hf(3)–$O_I(1)$ bonds (which are essentially perpendicular to the vertical polar axis) shrink upon in-plane compression. We have thus achieved the desired control on the atomic environment of the $O_I$ atoms. (The analogous results for

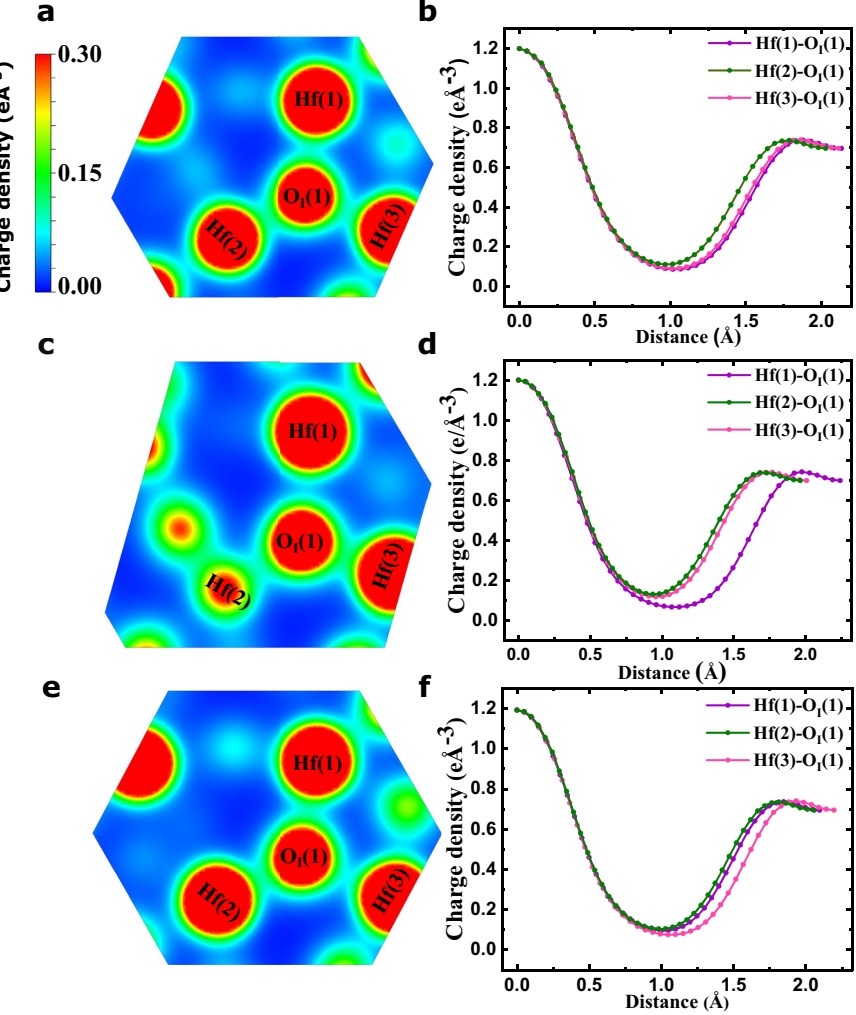

**Fig. 4 Visualizing Hf–O bonds.** Computed electronic charge density for the unperturbed ferroelectric phase of $HfO_2$ (panels **a** and **b**) as well as for the structures obtained at $\eta_{epi} = -7$ % (**c** and **d**) and $\eta_{epi} = +4$ % (**e** and **f**). Panels **a**, **c** and **e** show a contour plot of the charge density within a plane that approximately contains the $O_I(1)$ atom highlighted in Fig. 1 as well as its three nearest-neighboring Hf atoms. Panels **b**, **d** and **f** show the charge density along lines connecting the central oxygen with each of its three nearest-neighboring Hf cations. In panel **c**, the red globe at the top left of the Hf(2) atom corresponds to a neighboring oxygen anion that gets close to the shown plane.

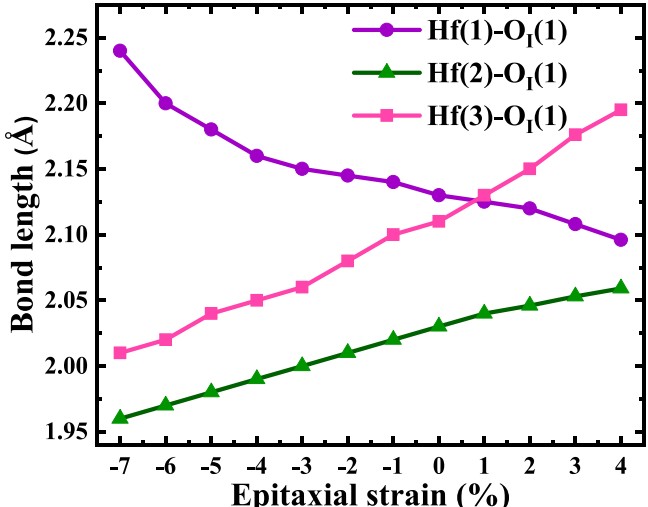

**Fig. 5 Controlling bonds with epitaxial strain.** Lengths of the Hf(1)-$O_I(1)$, Hf(2)-$O_I(1)$ and Hf(3)-$O_I(1)$ bonds defined in Fig. 1, computed as a function of epitaxial strain.

the Hf–O bonds of the $O_{II}$ atoms are shown in Supplementary Fig. 5.)

Figure 6 shows the evolution of the $e_{33}$ piezoresponse component, as a function of $\eta_{epi}$, obtained from DFPT calculations exactly as in the bulk case. We find that the frozen-ion contribution $\bar{e}_{33}$ remains nearly constant (and positive) in the whole range of strains. In contrast, the lattice-mediated part of the response (red line in the Figure) changes very markedly in a monotonic way. As a result, the total $e_{33}$ changes as well: it reaches its strongest negative response at tensile strains ($\eta_{epi} > 0$) and eventually switches to positive values as we compress the material in-plane!

Let us stress that this change of sign in $e_{33}$ occurs even though we have a positive $P_3 > 0$ for all considered $\eta_{epi}$ values. Indeed, as shown in Supplementary Fig. 6, we find that the polarization grows beyond 70 $\mu C\ cm^{-2}$ for epitaxial compression over $-5$ %, an evolution that is perfectly consistent with that of the structural distortions (Hf–$O_I(1)$ bonds in Fig. 5). Hence, the longitudinal piezoresponse changes sign even though the material remains in the same polar state; as far as we know, such an effect had never been observed (or predicted) before in a ferroelectric.

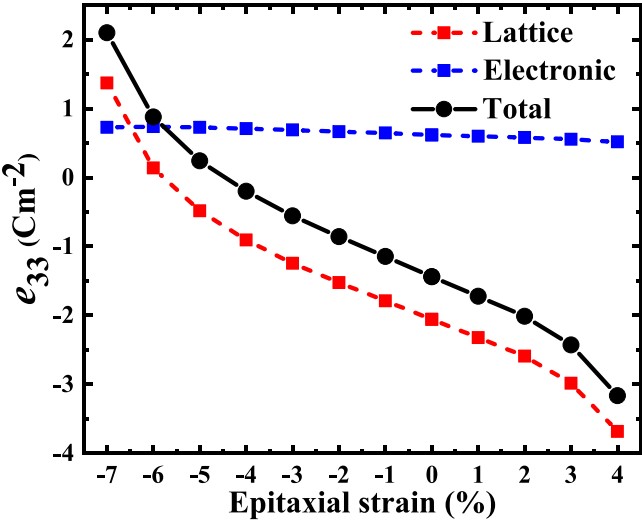

**Fig. 6 Computed $e_{33}$ piezoresponse component as a function of epitaxial strain.** The total $e_{33}$ (black) is split into frozen-ion (blue) and lattice-mediated (red) contributions.

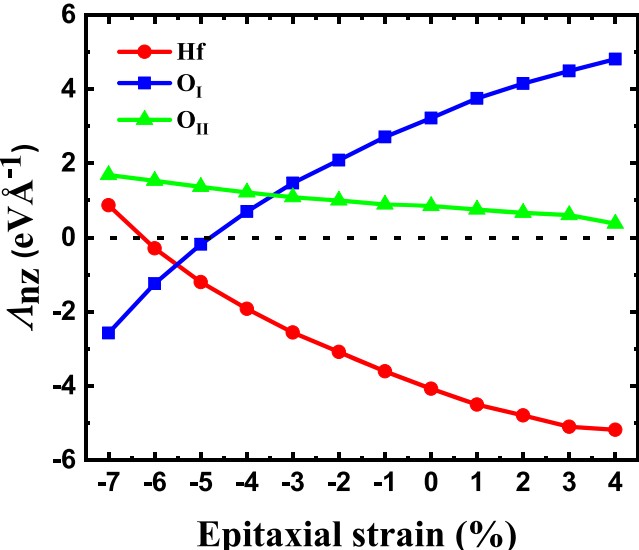

**Fig. 7 Epitaxial strain dependence of the $\Lambda$ components that control the $e_{33}$ response (see text).** More precisely, the shown components quantify the third (vertical) component of the force that acts on the Hf, $O_I$ and $O_{II}$ atoms as a consequence of an applied strain $\eta_3 > 0$.

We can easily track down the sign change of $e_{33}$ to the key components of the force-response internal-strain tensor, whose evolution with $\eta_{epi}$ is shown in Fig. 7. At zero strain, we have the situation already discussed above: when we stretch ($\eta_3 > 0$) the state with $P_3 > 0$, the system's response involves $O_I$ anions moving up (the corresponding $\Lambda_{nj}$ component is positive) and all Hf cations moving down (negative $\Lambda_{nj}$ component), which results in a reduction of the polarization $P_3$. Then, as we compress in-plane ($\eta_{epi} < 0$), the signs of the $O_I$ and Hf displacements eventually reverse, and so does the piezoresponse to $\eta_3$. Figure 7 also displays the key $\Lambda_{nj}$ component for the $O_{II}$ anions; the associated response—relatively small and largely independent of the epitaxial strain—does not play any role in the sign change of $e_{33}$.

Finally, and most importantly, we can check whether our physical picture for the sign of $e_{33}$ is correct. The key results are given in Fig. 4c–f, which shows the computed charge density as obtained for the limit cases with $\eta_{epi} = -7\%$ (panels c and d) and $\eta_{epi} = +4\%$ (panels e and f).

The result in Fig. 4c, d is particularly clear. For strong in-plane compression, the Hf(1)–$O_I$(1) bond is all but broken, as consistent with the long interatomic distance shown in Fig. 5. This suggests that, in this limit, the piezoresponse of the material will be controlled by the Hf(2)–$O_I$(1) and Hf(3)–$O_I$(1) links. Based on this assumption, we expect that a stretching along the vertical direction will result in the $O_I$(1) anion moving down and the Hf cations moving up, so that these two bond lengths change as little as possible. This is exactly what we find in our response calculations; and this behavior results in the obtained $e_{33} > 0$.

In the other limit ($\eta_{epi} = +4\%$, Fig. 4e, f) we find that the Hf(1)–$O_I$(1) and Hf(2)–$O_I$(1) bonds remain strong, while the Hf(3)–$O_I$(1) seems relatively weak. Hence, this case is similar to the bulk-like situation discussed above and corresponding to $\eta_{epi} = 0\%$. The only difference is that the preponderance of the vertical Hf(1)–$O_I$(1) bond can be expected to grow, which should result in a stronger $e_{33} < 0$, as we indeed obtain.

Hence, our epitaxial-strain calculations confirm that the proposed physical understanding of the longitudinal piezo-response of HfO$_2$ is essentially correct: the peculiar atomic environment of the active oxygen atoms, and the tendency to maintain the optimal length of the dominant Hf–$O_I$ bonds, determine the sign of $e_{33}$. Our calculations also show that the value of $e_{33}$ is strongly tunable, and can even change sign,

provided one is able to act upon said atomic environment. Epitaxial strain gives us a control knob to do this.

Let us conclude this part by noting that, as it is obvious from Fig. 6, our calculations predict that it is possible to find epitaxial conditions such that $e_{33} = 0$ despite the fact that HfO$_2$ remains polar with space group $Pca2_1$. This result may seem surprising, as the $Pca2_1$ symmetry allows for a non-zero $e_{33}$. However, let us recall that, by itself, symmetry does not tell us anything about the sign or magnitude of $e_{33}$; indeed, a continuous variation of the interatomic interactions responsible for the $e_{33}$ response can change its sign, and even make it vanish, as found here.

**Final remarks.** Our first-principles analysis thus reveals the atomistic reasons why the predicted longitudinal piezoelectric response of HfO$_2$ ($e_{33}$ or $d_{33}$) is negative. More specifically, we show that, when hafnia is strained along its polar axis, the material reacts by shifting the oxygen anions responsible for its spontaneous polarization, so as to best preserve the equilibrium distance of the corresponding Hf–O bonds. Naturally, this atomic rearrangement affects the polarization, in such a way that it grows when the strain is compressive, yielding a negative longitudinal effect.

Guided by this observation, we are able to identify a strategy to tune the piezoresponse –by controlling the chemical environment of the active oxygens–, showing that it can be enhanced or reduced, and even reversed to obtain a positive effect. Admittedly, the specific strategy tested here may not be applicable in practice. (We predict that large compressive epitaxial strains, beyond $-5\%$, are needed to change the sign of $e_{33}$.) Nevertheless, our qualitative result is important: to the best of our knowledge, this is the first example of a ferroelectric whose piezoelectric response can be reversed by a continuous modification of the lattice, without switching its polarization. This possibility is unheard of among ferroelectrics, and certainly inconceivable in perovskite oxides.

The theoretical prediction of a negative longitudinal piezo-response is a robust one, corroborated in several ways by us and also obtained by other authors[13,14]. Further, we are not aware of any instance where the theoretical sign of the piezoresponse (as predicted by first-principles methods based on DFT, like the ones

used here) contradicts the experimental observation. Hence, our experimental ratification of the negative effect –by means of a careful PFM investigation of two reference ferroelectrics (PZT and PVDF) as well as HfO$_2$, all treated in exactly the same way so that a direct comparison can be made– comes as no surprise. The reasonable quantitative agreement between the computed effect (about $-2.5$ pm V$^{-1}$) and the one estimated from experiments (between $-2$ pm V$^{-1}$ and $-5$ pm V$^{-1}$) further strengthens our confidence in the results presented here. Note that the results of ref. [35] on pristine Hf$_{0.5}$Zr$_{0.5}$O$_2$ samples also suggest a negative effect.

Having said this, it is important to recall that the vast majority of published experiments suggest a positive longitudinal piezo-response $d_{33,eff}$[11,32,33]. Indeed, in the course of this work, we found ourselves that the application of the same experimental protocol to other HfO$_2$ films (thicker, grown by different means) yields a positive longitudinal effect. Hence, we find that different HfO$_2$ samples may present $d_{33,eff}$ of different sign. This is a surprising observation, but one that resonates with our prediction that hafnia's piezoresponse can be reversed without switching its polarization. May the differences in the measured sign of $d_{33,eff}$ be related to that?

It is thus clear that the experimental question of piezoelectricity in HfO$_2$-based compounds is still open and full of promise. Additional studies will be needed to evaluate how various factors (processing conditions, chemical composition, thickness, mechanical boundary conditions, electrical cycling) affect the outcome. We must try to correlate specific results for $d_{33,eff}$ with specific (structural) features in the corresponding samples, distinguishing between intrinsic and extrinsic contributions to the response, a task for which first-principles theory may prove a valuable aid to experiment. This is a most appealing challenge, from both fundamental and applied perspectives. On the one hand, it may allow us to understand and master unprecedented ways to control piezoelectricity in ferro-electrics. On the other hand, it may allow us to optimize piezoelectricity in HfO$_2$ up to the point required for applications. We hope the present work will bring impetus to this effort.

## Methods

**First-principles simulations**. Our calculations are carried out using first-principles DFT as implemented in the Vienna Ab-initio Simulation Package (VASP)[39,40]. We employ the Perdew-Burke-Ernzerhof formulation for solids (PBEsol)[41] of the generalized gradient approximation for the exchange-correlation functional. In our calculations, the atomic cores are treated within the projector-augmented wave approach[42], considering the following states explicitly: $5d$, $6s$, $6p$ for Pb; $3p$, $4s$, $3d$ for Ti; $2s$, $2p$ for O; and $5s$, $5p$, $6s$, $5d$ for Hf. To calculate the response functions we use DFPT[22]. All the calculations (for both PbTiO$_3$ and HfO$_2$) are carried out using a plane-wave energy cutoff of 600 eV. A $6 \times 6 \times 6$ **k**-point sampling of the Brillouin zone[43] is used for PbTiO$_3$ (corresponding to a 5-atom unit cell), while for HfO$_2$ we use a $4 \times 4 \times 4$ grid (corresponding to a 12-atom unit cell). The structures are fully relaxed until the residual forces fall below 0.01 eV Å$^{-1}$ and residual stresses fall below 0.1 GPa. We checked that these calculation conditions yield well-converged results.

To verify our predictions for the piezoelectric properties of HfO$_2$, we also run analogous DFPT calculations using the ABINIT first-principles package[44]. In this case, we also consider the Perdew-Burke-Ernzerhof formulation for solids of the generalized gradient approximation for the exchange-correlation functional. We use scalar relativistic norm-conserving Vanderbilt pseudopotentials as implemented in the ABINIT package[45]. In the calculations we treat explicitly the semicore states of Hf ($5s$, $5p$, $4f$, $5d$ and $6s$) and O ($2s$). We consider a plane-wave cutoff energy of 60 hartree and a $4 \times 4 \times 4$ **k**-point sampling of the Brillouin zone. We relax the structures until the residual forces fall below $10^{-6}$ hartree bohr$^{-1}$.

**Sample preparation**. 200 nm-thick (111)-oriented PZT films with a Zr/Ti ratio of 40/60 were fabricated by magnetron sputtering on the Pt bottom electrode. Reactive ion etching was carried out to fabricate capacitors with 50 nm thick IrO$_2$ top electrodes with lateral dimensions of $80 \times 80$ μm$^2$.

The 20 nm-thick La:HfO$_2$ films were grown by atomic layer deposition on TiN bottom electrodes and capped with a TiN top electrode. The TiN/La:HfO$_2$/TiN stack was then annealed in a N$_2$ atmosphere at 800 °C for 20 s. Details of the growth process for the La:HfO$_2$ films can be found in ref. [46].

The 12 monolayer-thick (21.6 nm) PVDF films were deposited on Pt/Si substrates by Langmuir-Blodgett methods[16].

**Sample characterization**. Switching spectroscopy PFM measurements were performed on a commercial atomic-force-microscopy system (MFP-3D, Asylum Research) in the resonance tracking mode using single-crystalline diamond tips (D80, K-Tek, Nanotechnology) and Pt-coated tips (HQ:DPER-XSC11, Mikro-Masch). Electrical bias was applied to the top electrode using an external probe, with the frequency of the ac modulation signal around 350 kHz and 650 kHz for the D80 and the Pt-coated tips, respectively. For the PVDF thin films, the conducting tip acted as a local top electrode.

In the DART mode, a feedback loop tracks a shift in the resonance frequency by measuring the difference in the PFM amplitudes for the two drive signals—above and below the resonance frequency. The PFM loops shown in this work were all obtained 3 kHz below the resonance frequency.

## Data availability

All relevant data are included in the figures and tables in the manuscript and in the Supplementary Information file. Additional data (if any) are available from the authors upon reasonable request.

## Code availability

The first-principles simulations were done with VASP, which is proprietary software for which the LIST group owns a license, and ABINIT, which is available at https://www.abinit.org under the GNU General Public License.

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

## Acknowledgements

This work was funded by the Luxembourg National Research Fund (FNR) through grants PRIDE/15/10935404 "MASSENA" (S.D. and J.Í.) and FNR/C18/MS/12705883 "REFOX" (H.A. and J.Í.). Work at UNL was supported by the National Science Foundation through EPMD (Grant No. ECCS-1917635) Programs. Work at Namlab was financially supported out of the State budget approved by the delegates of the Saxon State Parliament. We thank Alfred Kersch for insightful comments on the manuscript.

## Author contributions

S.D. performed the first-principles study, assisted by H.A. and supervised by J.Í. Samples were prepared by S.G., E.D, C.R., and U.S., and the PFM characterization was carried out by P.B., H.L. and A.G. The manuscript was written by J.Í., S.D., A.G., and S.G., with contributions from H.A., U.S., H.L., P.B. and E.D. J.Í. conceived and coordinated the work.

## Competing interests

The authors declare no competing interests.
