## [Peer Review File · Nature Communications]

REVIEWER COMMENTS

Reviewer #1 (Remarks to the Author):

The work by S. Dutta et al. tackles the emergent subject of negative longitudinal piezoelectric response in ferroelectric HfO₂ using first-principles calculations and related PFM experiments. The topic is technically relevant and the manuscript has been nicely prepared. This referee's otherwise questions and remarks are given below.

1. Through the authors' thorough theoretical investigations, the negative piezoelectric response arises from the cooperative displacements of O(I) and Hf, the latter of which is numbered in Hf(1), Hf(2), and Hf(3). In the ferroelectric polar phase of HfO₂ (space group, Pca21), there is only one Hf crystallographic site (Table S1) and each O(1) is bound to three nearest-neighbor Hf. The origin of the negative piezoelectric response is derived from this Hf-O(1) sublattice argument and this referee found the evidences provided (Figs. 6-8 and Figs. S4-S5) consistent. However, the notation of Hf(1), Hf(2), and Hf(3) could lead to the confusion of unspecified Hf crystallographic sites and it is advised to improve this part by a more explicit explanation on the sublattice conception. Moreover, the argued atomic displacements of O(I) and Hf are intimately correlated with the polar 21-symmetry element along c-axis. A more insightful discussion on the symmetry linking between the atomic displacements and the characteristic space group could provide the readers with potential hints on future functionality optimizations and also otherwise material searches.

2. In the investigation of the negative piezoelectric response by the simulation of applied isotropic strains in ab-plane, it is surprising to this referee that the displacement characteristic of other oxygen, O(II), has not been mentioned, let alone a further discussion. It would be difficult for this referee to believe that strains only impact the atomic sites of O(I) and Hf, but not O(II). This is the very unclear part of this work and definitely needs to be improved.

If the authors can satisfactorily improve the text (above #1) and unambiguously clarify the role of O(II) in the negative piezoelectric response of HfO₂ (above #2), this referee would be happy to suggest an active consideration of the work in the journal.

Reviewer #2 (Remarks to the Author):

The authors reported a combined theoretical and experimental analysis of the negative longitudinal piezoelectricity in HfO₂. They show that in the orthorhombic HfO₂, the bond length of O(1) with Hf(1) (nearly in polar direction) dominates among the three nearest Hf atoms in the application of the strain along the polar axis. This bond length decreases when the strain is positive while increases in negative strain. They tried to explain this exotic behavior by calculating the bond length, charge density, force constant (Λ) which control the piezo-response component (e_{33}) and e_{33} as a function of epitaxial strain. They also compare their results with PbTiO₃ which shows a conventional positive piezoelectricity. Although this is a very timely interesting report in the ferroelectric history, the previous papers (Ref. [11,12]) have already reported the same negative piezoelectricity. Therefore, the explanation of the origin of the negative piezo. should be strengthened for the paper to be published.

Major

1. Their major results (negative longitudinal piezoelectricity in HfO₂) have been already reported in Ref. [11] (Phys. Rev. Applied 12, 034032 (2019)) and Ref. [12] (Phys. Rev. Lett. 125, 197601 (2020)). This paper compares HfO₂ stress tensor and PbTiO₃ stress tensor to show their opposite cation-anion movement under the strain, which yield opposite d_{33} sign each other. However, the authors admit that it does not provide fully satisfying physical understanding. Of course, a stronger bond such as O(1)-Hf(1) by itself does not guarantee a negative piezoelectricity. Probably, a crystal symmetry of HfO₂ adds up to the effect. Would the authors provide a more intensive comparison with perovskites such as PbTiO₃? (In PbTiO₃, the stronger bond between Ti and apical oxygen drives a POSITIVE d_{33} . Then why the stronger bond between Hf(1) and O drives a NEGATIVE d_{33} in HfO₂?)

Minors.

1. In Fig.9 (a), the authors did not tell about the red spot near to Hf(2).
2. Why the d_{33} values of the single-crystal HfO₂ (-2.54 pm/V) and of the DFT (-0.94 pm/V) differ significantly?

Reviewer #3 (Remarks to the Author):

The work Dutta et al reports on piezoelectricity in HfO₂ discussing on the recently predicted negative longitudinal piezoelectric effect. The authors use first principles calculations and PFM technique for supporting their claim. Particularly, the paper shows how the chemical coordination of the oxygen atoms can induce negative longitudinal piezoelectric effect in HfO₂, thus suggesting a pathway for controlling the ferroic response of HfO₂ by controlling the strain environment of the ferroelectric layer. The paper is well written and discusses a topic of potential interest. However, the report is considered a quite specialistic simulation work and, as such, the referee does not believe that Nature Communication is the best venue for publication. In addition, to the referee's knowledge, multiple reports have already addressed various limitations of resonant and non-resonant PFM methods (including DART) for the quantitative analysis of HfO₂. As discussed by the authors in their text, the polycrystalline nature of the film with the presence of grains with multiple orientation, defects, traps, and grain boundaries generally complicates the interpretation of PV loops for HfO₂-based materials. Therefore, the experimental evidence provided in the text could not be considered definitive in the referee's opinion.

Let us start by thanking the reviewers for their careful consideration of our manuscript and constructive comments. In the following we provide detailed responses to all of them.

For convenience, all the changes made in the manuscript are marked in red.

Comments from Reviewer #1

The reviewer expresses a positive opinion of our work, as he/she writes:

The work by S. Dutta et al. tackles the emergent subject of negative longitudinal piezoelectric response in ferroelectric HfO₂ using first-principles calculations and related PFM experiments. The topic is technically relevant and the manuscript has been nicely prepared. This referee's otherwise questions and remarks are given below.

Then, the reviewer makes the following specific remarks.

Comment #1.1

1. Through the authors' thorough theoretical investigations, the negative piezoelectric response arises from the cooperative displacements of O(1) and Hf, the latter of which is numbered in Hf(1), Hf(2), and Hf(3). In the ferroelectric polar phase of HfO₂ (space group, Pca21), there is only one Hf crystallographic site (Table S1) and each O(1) is bound to three nearest-neighbor Hf. The origin of the negative piezoelectric response is derived from this Hf-O(1) sublattice argument and this referee found the evidences provided (Figs. 6-8 and Figs. S4-S5) consistent. However, the notation of Hf(1), Hf(2), and Hf(3) could lead to the confusion of unspecified Hf crystallographic sites and it is advised to improve this part by a more explicit explanation on the sublattice conception.

We are glad to learn that the reviewer found our evidence and arguments convincing.

We agree our original notation can be improved. In the original version of the article, we used numbers in parenthesis to label both the sublattices of symmetry-equivalent atoms (e.g., "O(1)") and the atoms in a certain sublattice (e.g., "Hf(1)"). This may indeed be confusing.

In the revised version we adopt a different notation that we hope will be clearer. We denote the sublattices of symmetry-equivalent atoms by a Roman number that appears as a subindex. In contrast, we label atoms belonging to a particular sublattice by an Arabic number in parenthesis. Hence, for example, for the two sublattices of oxygens in the Pca2₁ phase we use O_I and O_{II}, respectively. Then, “O_I(1)” refers to a particular oxygen atom (number 1) belonging to the first oxygen sublattice (number I). Finally, as we have only one Hf sublattice in the Pca2₁ phase, we simply use “Hf(1)”, “Hf(2)”, etc. to label individual atoms.

We explicitly explain (and systematically apply) this notation in the revised manuscript. See the discussion about Figure 1 and Table S1, at the start of the “Basic first-principles predictions” section.

Comment #1.2

Moreover, the argued atomic displacements of O(I) and Hf are intimately correlated with the polar 21-symmetry element along c-axis. A more insightful discussion on the symmetry linking between the atomic displacements and the characteristic space group could provide the readers with potential hints on future functionality optimizations and also otherwise material searches.

We are afraid we do not understand what kind of insightful discussion the reviewer would like us to provide here. Yet, we do have some comments to make about using symmetry for materials discovery or optimization.

It is true that symmetry analysis may help discover new materials with particular properties, a good example being the identification of new ferroelectrics by systematic pseudo-symmetry search [see, e.g., Kroumova et al., Acta Cryst. B 58, 921 (2005)]. However, we would argue that one *cannot* use symmetry as a guide for further property optimization. Let us give two arguments supporting this point:

- The results in our paper show that symmetry says nothing about the nature of the piezoelectric response of HfO₂ or how to optimize it. Recall our calculations for the variation of ϵ_{33} as a function of epitaxial strain. The symmetry of the ferroelectric

phase ($Pca2_1$) is the same throughout the considered strain range; and the symmetry characteristics of the O_1 and Hf atom sublattices also remain the same (i.e., the corresponding Wyckoff positions do not change). However, the epitaxial strain affects the relative positions of the atoms, and this in turn results in a big change of e_{33} , which shifts from negative (for null epitaxial strain) to positive (under sufficient epitaxial compression). In fact, for a “critical” value of the epitaxial compression we have $e_{33}=0$, even though the space group remains $Pca2_1$. This null response is not mandated by symmetry, but stems from a cancellation of rather-intricate interactions that change continuously as we vary the epitaxial strain. This is clear evidence that it is the magnitude of the atomic interactions and their interplay, not the symmetry, what allows us to control or optimize e_{33} .

- Let us also recall the materials with the largest piezoelectric response: perovskite oxide solid solutions like $PbZr_{1-x}Ti_xO_3$ (PZT). Compounds like PZT present record values of the piezoelectric and dielectric response in the *morphotropic* region of their composition—temperature phase diagram, i.e., at the boundary between two ferroelectric states of different symmetries, rhombohedral and tetragonal in the case of PZT. In the transition region, the polarization evolves progressively between the rhombohedral and tetragonal states, and the associated energy landscape is very flat. This flatness implies that changing the polarization has a negligible energy cost, and we thus have large responses.

In PZT, the symmetry in this morphotropic phase is monoclinic, a direct result of the progressive rotation of the polarization between the rhombohedral and tetragonal states. And it is true that, in perovskite solid solutions like PZT, monoclinic phases can be taken as a guarantee for large responses, because (as far as we know) they always appear associated to a similar progressive rotation of the polarization and an attendant flat energy landscape. However, by itself, a monoclinic symmetry does *not* guarantee a flat energy landscape in other materials families. Why should it? The fact is that symmetry tells us which effects are allowed to exist but, by itself, implies nothing about their magnitude.

For these reasons, we think it is wrong to propose symmetry as a criterion or guide for the search of materials with optimum properties, piezoelectric in this case. Hence, we have not made any change in the manuscript concerning this point.

Nevertheless, in the revised version we comment briefly on the fact that we predict there exist epitaxial conditions at which $e_{33}=0$ although the symmetry remains $Pca2_1$. We hope this will convey the message that symmetry does not tell us anything about the sign or magnitude of e_{33} . (See new last paragraph of the “Prediction of a tunable piezoresponse” section.)

Comment #1.3

2. In the investigation of the negative piezoelectric response by the simulation of applied isotropic strains in ab-plane, it is surprising to this referee that the displacement characteristic of other oxygen, O(II), has not been mentioned, let alone a further discussion. It would be difficult for this referee to believe that strains only impact the atomic sites of O(I) and Hf, but not O(II). This is the very unclear part of this work and definitely needs to be improved.

It is true that the epitaxial strain also impacts the positions of the O(II) atom sublattice (O_{II} in the revised notation introduced in comment #1.1). However, our results indicate that the O_{II} atoms never have a relevant contribution to the longitudinal piezoelectric response, which is why we did not mention them in the original version of our manuscript.

To make this point explicit and avoid any doubts, we have implemented the following changes: (1) We have included a new Supplementary Figure S5 showing how the distances between a representative O_{II} atom and its neighboring Hf atoms evolve as a function of epitaxial strain. (2) In Fig. 7 of the main text (previously Fig. 8), we have added the relevant Λ_{nz} component contributing to e_{33} and corresponding to a representative O_{II} atom, showing that it remains relatively inert (as compared to those for Hf and O_I) in the considered range of epitaxial strain. (3) We have added a brief comment on this point in the text.

Comments from Reviewer #2

The reviewer starts by making a general assessment of our manuscript...

The authors reported a combined theoretical and experimental analysis of the negative longitudinal piezoelectricity in HfO₂. They show that in the orthorhombic HfO₂, the bond length of O(1) with Hf(1) (nearly in polar direction) dominates among the three nearest Hf atoms in the application of the strain along the polar axis. This bond length decreases when the strain is positive while increases in negative strain. They tried to explain this exotic behavior by calculating the bond length, charge density, force constant (Λ) which control the piezo-response component (e_{33}) and e_{33} as a function of epitaxial strain. They also compare their results with PbTiO₃ which shows a conventional positive piezoelectricity. Although this is a very timely interesting report in the ferroelectric history, the previous papers (Ref. [11,12]) have already reported the same negative piezoelectricity. Therefore, the explanation of the origin of the negative piezo. should be strengthened for the paper to be published.

... where he/she indicates that the novelty of our work resides, essentially, on the explanation of the negative piezoelectric response, which he/she thinks needs to be reinforced.

We are more than happy to take advantage of the reviewer's specific suggestions to strengthen our proposed physical interpretation (see below). However, before that, we would like to point out that the novelty of our work is *not* limited to these explanations.

Let us stress that in this work we provide a careful experimental confirmation of the negative linear piezoelectric effect, and show evidence for its intrinsic nature. This is a first in the literature, and therefore an important and novel result of our work.

Further, based on our new physical insights, we identify a strategy to revert the sign of the linear piezoelectric response via a continuous modification of the ferroelectric phase of HfO₂ (using epitaxial strain). As far as we can tell, such a behavior has never been predicted or experimentally observed before, and most experts in the field (including us)

would probably deem it impossible. Hence, this constitutes a novel and very surprising result of our work.

Having said this, we now address the specific comments made by the reviewer.

Comment #2.1

Major

1. Their major results (negative longitudinal piezoelectricity in HfO₂) have been already reported in Ref. [11] (Phys. Rev. Applied 12, 034032 (2019)) and Ref. [12] (Phys. Rev. Lett. 125, 197601 (2020)). This paper compares HfO₂ stress tensor and PbTiO₃ stress tensor to show their opposite cation-anion movement under the strain, which yield opposite d₃₃ sign each other. However, the authors admit that it does not provide fully satisfying physical understanding. Of course, a stronger bond such as O(1)-Hf(1) by itself does not guarantee a negative piezoelectricity. Probably, a crystal symmetry of HfO₂ adds up to the effect. Would the authors provide a more intensive comparison with perovskites such as PbTiO₃? (In PbTiO₃, the stronger bond between Ti and apical oxygen drives a POSITIVE d₃₃. Then why the stronger bond between Hf(1) and O drives a NEGATIVE d₃₃ in HfO₂?)

Let us restate here some of the messages in our article, to avoid misunderstandings. We do believe that a mere description in terms of the tensors involved in the piezoelectric effect (Born charges, force-constant matrix, force-response internal strain, and elastic constants) does *not* provide a satisfying physical picture of why $e_{33} < 0$ in HfO₂. That's why we kept analyzing our results until we identified how particular Hf—O bonds control the response. We believe that this picture of “strong bonds” wanting to keep their preferred length – which is further supported by our calculations on how to control the sign of e_{33} – does provide a satisfying physical interpretation.

Now, the reviewer poses an apparent paradox: how can it be that our “strong bond” picture yields a positive d₃₃ in PbTiO₃ but a negative d₃₃ in HfO₂? Further, the reviewer wonders about a symmetry-based explanation to this apparent paradox. Let us make two points about this:

- As already mentioned in response to comment #1.2 above, symmetry cannot explain the sign of a piezoelectric response. Indeed, our calculations for epitaxially-strained HfO₂ show that we can have exactly the same symmetry (Pca2₁) but obtain e_{33} positive or negative depending on other factors (i.e., the changes in the Hf—O bond hierarchy driven by epitaxial compressive strain). This strongly suggests that any symmetry-based solution to the paradox posed by the reviewer is likely to be wrong.
- The key difference between PbTiO₃ and HfO₂ is the coordination of the piezoelectrically active ions, which is very different between these two compounds. This was already described in our original manuscript, but we agree it is possible to provide a deeper insight into the origin of the negative e_{33} of HfO₂. We do so in the following.

Let us first recall the case of PbTiO₃, which is relatively simple. The Ti—O interactions are key to the occurrence of ferroelectricity in this material. (Along with Pb—O interactions not discussed here for simplicity.) As shown in Fig. 3(a—c) of our manuscript (also in Fig. R1(a—c) of this response, for convenience), the onset of ferroelectricity in PbTiO₃ consists, in essence, in the upward movement of the central Ti atom and downward movement of the O anions, which results in a positive polarization along the c (vertical) direction. As a result of this polar distortion, a shorter (strong) Ti—O bond is formed, and this bond controls the e_{33} response of the polar phase. More specifically, if we stretch the cell along the polar axis ($\eta_3 > 0$), the Ti and O atoms react by approaching each other and thus keep their preferred distance. This “approaching” involves an upward shift of the Ti cation and a downward movement of the apical oxygen (Fig. 3(c)), resulting in an increased polarization and a positive e_{33} .

Figure R1. Sketches showing the relationship between the paraelectric and ferroelectric states of PbTiO_3 (panels (a) and (b), respectively) and HfO_2 (panels (d) and (e), respectively), as well as their response to a positive tensile strain along the polar axis (panels (c) and (f), respectively). The arrows in panels (b) and (e) represent the atomic displacement connecting the paraelectric and ferroelectric states. The arrow in panels (c) and (f) stand for the ionic motions that occur in response to the applied strain, as obtained from first principles. This figure is the new Fig. 3 of the manuscript, whose caption can be consulted for further details.

HfO_2 is different in that the anion that is piezoelectrically active (the highlighted $\text{O}_i(1)$ ion in Fig. 1(b) of our manuscript (previously Fig. 4)) is strongly bonded to not one, but three Hf cations (labeled Hf(1), Hf(2) and Hf(3) in Fig. 1(b) of our manuscript). However, from these three cations, only one (Hf(1)) forms a Hf—O bond that is approximately parallel to the polar axis, while the other two cations (Hf(2) and Hf(3)) form Hf—O bonds that are essentially perpendicular to the polarization direction.

Let us now consider Figs. R1(d) and R1(e) of this response letter. It is clear from those figures that, as we move from the paraelectric to the ferroelectric state of HfO_2 , the oxygen of interest ($\text{O}_i(1)$) shifts down and to the right, mainly approaching the Hf(2) cation below it. As a result, in the polar phase we have a short Hf(2)— $\text{O}_i(1)$ bond (2.03

Å; see Fig. 1(b)) while Hf(3)—O_I(1) and Hf(1)—O_I(1) are a bit longer (2.11 Å and 2.13 Å, respectively). (For comparison: in the paraelectric cubic phase, all O atoms have 4 nearest-neighboring Hf atoms, and all the corresponding symmetry-equivalent Hf—O bonds are 2.17 Å long; see the revised Fig. 1(a) of our manuscript.) Hence, in HfO₂, the Hf—O bond that is best aligned with the polarization direction (i.e., the Hf(1)—O_I(1) pair) is *not* the main driving force for the ferroelectric state to occur. This is critically different from the case of PbTiO₃.

Note that this situation is consistent with recent discoveries on the character of ferroelectricity in HfO₂. From first-principles studies of the phonons of HfO₂'s polymorphs (see, e.g., Phys. Rev. B 65, 233106 (2002) and Phys. Rev. Mats. 5, 064405 (2021)), we know that there is no native ferroelectric instability of the cubic phase, and that the occurrence of the Pca21 polymorph depends on the prior condensation of a non-polar mode that connects the cubic state (taken as a convenient reference in our manuscript) with a well-known tetragonal paraelectric polymorph (not discussed here). Thus, taking the cubic phase as a reference, HfO₂ is *not* a proper ferroelectric, and we have no reason to expect it to behave as the well-known proper-ferroelectric perovskites, like PbTiO₃.¹

Now, imagine we stretch the HfO₂ cell along the polarization direction (Fig. R1(f)). The Hf(1)—O_I(1) pair will be the one most affected by this strain, as this bond lies essentially along the polar axis. Hence, the O_I(1) anion will react to keep the preferred Hf(1)—O_I(1) bond distance. More precisely, O_I(1) will move up, as indeed obtained from our first-principles calculations and indicated in Fig. R1(f). This shift goes *against* the polar distortion connecting the paraelectric (Fig. R1(d)) and ferroelectric (Fig. R1(e)) phases. It clearly results in a negative contribution to the polarization and, hence, we have $\epsilon_{33} < 0$.

We have significantly revised and expanded the relevant discussion in our manuscript to include this point. We have also replaced the original Fig. 3 (devoted only to PbTiO₃) with Fig. R1 in this response, which shows a side-by-side comparison of PbTiO₃ and HfO₂ that we hope will be illuminating. Additional changes are made throughout the text (for

¹ If we were to take the tetragonal phase as a centrosymmetric reference, it would still be tricky to describe HfO₂ as a regular proper ferroelectric, because of the prominent role played by a strong trilinear coupling between the weak polar instability and two hard modes (see Phys. Rev. Mats. 5, 064405 (2021)). This discussion is beyond the scope of the present work, and we do not address it here.

example, the original Figs. 1 and 4 are merged into the new Fig. 1; two new references are added) to accommodate the new discussion. We do hope this will make our explanations more insightful and convincing.

Comment #2.2

1. In Fig.9 (a), the authors did not tell about the red spot near to Hf(2).

That red spot is associated to the electronic cloud around a nearby oxygen that does not bind to Hf(2) and is thus irrelevant for the present discussion. Yet, we do agree this may be confusing. We have added a brief explanation in the caption of Fig. 8 (previously Fig. 9).

Comment #2.3

2. Why the d_{33} values of the single-crystal HfO₂ (-2.54 pm/V) and of the DFT (-0.94 pm/V) differ significantly?

We are not sure we understand this comment. Maybe our text was not clear enough, and the reviewer has misunderstood our discussion?

Let us start by noting that, as far as we know, there is no measurement of d_{33} in a single crystal of HfO₂. In fact, HfO₂ single crystals are exceedingly rare; in fact, the first single-crystal results were reported only a few months ago (see Nat. Mats. 20, 826 (2021)). Hence, in our manuscript we do not cite experimental measurements of piezoelectricity in HfO₂ single crystals, as we are not aware of any.

In our paper we do mention the *predicted* value of d_{33} for ideal monocrystalline HfO₂, as obtained from DFT. This value is -2.51 pm/V. We also indicate that, if we use our DFT data to try to approximate d_{33} of a polycrystalline sample with a perfectly random orientation of the grains (Eq. (3) of our manuscript), one gets -0.94 pm/V. Hence, these are both

theoretical predictions, which correspond to distinct limits: single crystal vs. untextured polycrystal.

Finally, as discussed in the “Experimental confirmation” part of our manuscript, our experimental measurements (of polycrystalline samples) suggest that d_{33} is between -2 pm/V and -5 pm/V. We make the point that these values are consistent with our DFT results.

About this last point, one should note that our calculations are done in the limit of 0 K, while our experiments are at room temperature. Further, the simulated idealized HfO₂ (monocrystalline, free of defects) is quite different from the real samples (polycrystalline, Y-doped, probably with a significant amount of point defects). Given these circumstances, we believe the correspondence between calculated and measured values is as good as can be possibly expected to be.

In our revised manuscript, at the end of the “Basic first-principles predictions” subsection, we have added an explicit statement about our theoretical predictions for d_{33} , citing the single-crystal and polycrystalline limits. We hope this will minimize the chances of misunderstanding.

Comments from Reviewer #3

The reviewer starts his/her report by praising our manuscript..

The work Dutta et al reports on piezoelectricity in HfO₂ discussing on the recently predicted negative longitudinal piezoelectric effect. The authors use first principles calculations and PFM technique for supporting their claim. Particularly, the paper shows how the chemical coordination of the oxygen atoms can induce negative longitudinal piezoelectric effect in HfO₂, thus suggesting a pathway for controlling the ferroic response of HfO₂ by controlling the strain environment of the ferroelectric layer. The paper is well written and discusses a topic of potential interest. However, the report is considered a quite specialistic simulation work and, as such, the referee does not believe that Nature Communication is the best venue for publication.

... but also indicates that, in his/her opinion, our work is for simulation specialists, not appropriate for the general readership of Nature Communications. We are very surprised by this comment and must respectfully disagree.

First, our work contains novel and important results at both the theoretical and experimental levels, results that complement and corroborate each other. We are not aware of any specialized simulation papers combining theory and experiment as we do in ours.

Second, our theoretical discussion is focused on the physical understanding of the negative longitudinal piezoresponse of HfO₂. Further, we discuss and explain the striking prediction of a novel physical effect, namely, how to control the sign of the piezoresponse via epitaxial strain. These discoveries are of broad interest to the community of ferroelectrics and functional oxides, and to researchers interested in HfO₂, including those working on field-effect transistors.

Third, all our results are presented in a pedagogical manner, offering novel insights into piezoelectricity in general (by means of the comparison with PbTiO₃) and the behavior of HfO₂ in particular. And our text is fully focused on the physical effects and their underpinnings, the technicalities of the calculations being absent from the main text.

We hope these arguments may encourage the reviewer to reconsider his/her criticism of our manuscript.

Then, the reviewer makes other specific comments.

Comment #3.1

In addition, to the referee's knowledge, multiple reports have already addressed various limitations of resonant and non-resonant PFM methods (including DART) for the quantitative analysis of HfO₂.

Although this is a valid concern, we would like to point out that PFM has already been successfully applied to acquire quantitative information on HfO₂-based ferroelectrics such as domain switching dynamics [i,ii], quantitative estimation of the longitudinal piezoelectric coefficient [iii,iv], etc. These examples confirm the veracity of the quantitative insights afforded by applying PFM to HfO₂-based ferroelectrics. If the reviewer has a specific concern regarding the specific set of the experimental data, we would appreciate it if it would be expressed more clearly. Otherwise, it appears that the reviewer dismisses the experimental results without any reason.

We would like to make a point that the measurement protocols used in our studies minimized any potential parasitic contributions to the PFM signal. As explained in detail in Section S1 in the Supporting Information, the PFM amplitude, A , is proportional to the longitudinal piezoelectric coefficient, $d_{33,eff}$ via the following relation:

$$A = d_{33,eff} V_0 Q \quad (1)$$

where V_0 is the driving ac voltage amplitude and Q is the quality factor associated with the cantilever dynamics at resonance [v]. The $d_{33,eff}$ of an unknown material can be obtained by comparing the PFM amplitude of the unknown material with that of a reference material with a known $d_{33,eff}$. In our measurements, IrO₂/PZT/Pt capacitors were used as the reference material. The $d_{33,eff}$ of the PZT capacitors was first obtained from quasi-static strain loop measurements by measuring the static deflection of the AFM cantilever during the application of a triangular voltage sweep at 1 Hz. The obtained

deflection signal could be then converted to the actual displacement of the cantilever through calibration of the optical lever sensitivity using force-distance curves. From the slope of the quasi-static strain loops, the $d_{33,eff}$ for the IrO₂/PZT/Pt capacitors was found to be 48 pm/V, in agreement with values reported in the literature [vi]. This value was then used to obtain the quality factor of the cantilever using Equation 1. The same cantilever was then used to measure the PFM hysteresis loops in the TiN/La:HfO₂/TiN capacitors. The $d_{33,eff}$ was then calculated for the TiN/La:HfO₂/TiN capacitors from Equation 1 using the Q factor obtained in the previous step. The obtained $d_{33,eff}$ values of around 2 pm/V to 5 pm/V in the TiN/La:HfO₂/TiN capacitors match very well with the values reported by interferometric methods [vii, viii, ix, x]. We agree that the obtained $d_{33,eff}$ are order of magnitude estimates due to the error inherent in the estimation of the Q factor; however, the excellent agreement of our obtained results with that reported in the literature via direct interferometric methods strongly supports the validity of our measurement protocol for the quantitative estimation of the $d_{33,eff}$.

On the other hand, the sign of the $d_{33,eff}$ determined from the phase signal in the PFM hysteresis loop measurements is related to the underlying material piezoelectric properties via the following relation between the strain generated in the out of plane direction, η_3 , due to the applied electric field in this direction, E_3 , [xi]:

$$\eta_3 = d_{33,eff} E_3 \quad (2)$$

In a material with a positive $d_{33,eff}$, such as PZT, the sample will expand (contract) when the applied field and the polarization are in the same (opposite) direction. In PFM, where a small ac drive voltage is used to probe the sample deformation via lock-in techniques, there can be an unknown instrument-related parasitic phase offset that can lead to an erroneous interpretation of the PFM phase signal with regards to the sign of the $d_{33,eff}$. This parasitic phase offset was identified using two approaches – by using a reference sample with a known $d_{33,eff}$ value and using the differential signal between the bias-on and the bias-off piezoresponse loops – allowing us to unambiguously determine the actual PFM phase related to the true piezoelectric deformation of the sample. The sample deformation and the ac driving signal will be in-phase (180° out of phase) when the polarization is pointing downwards (upwards) in a material with positive $d_{33,eff}$ and vice-versa for a material with a negative $d_{33,eff}$. This is the exact scenario observed in the PFM

phase loops obtained on known positive and negative $d_{33,eff}$ materials – PZT [xii] and PVDF [xiii, xiv], respectively, as shown in Figure R2 (a,c) – after identification of the parasitic phase offset. Furthermore, the agreement of the PFM phase signals with the corresponding quasi-static strain loops in PZT and PVDF (Figure R2 (b,d)) clearly highlight the validity of our measurement protocols. In conclusion, an unambiguous interpretation of the PFM signals can be and was obtained in this paper through careful consideration of the potential parasitic contributions to the PFM signal.

Figure R2: (a,b) PFM phase (top panel) and amplitude (bottom panel) loops (a) and quasi-static strain loop (b) measured in IrO₂/PZT/Pt capacitors where PZT has a positive $d_{33,eff}$. (c,d) PFM phase (top panel) and amplitude (bottom panel) loops (c) and quasi-static strain loop (d) measured in PVDF thin film where PVDF has a negative $d_{33,eff}$.

Comment #3.2

As discussed by the authors in their text, the polycrystalline nature of the film with the presence of grains with multiple orientation, defects, traps, and grain boundaries generally complicates the interpretation of PV loops for HfO₂-based materials. Therefore, the experimental evidence provided in the text could not be considered definitive in the referee's opinion.

We would like to point out that, in spite of the polycrystalline nature of the HfO₂ films, the local PFM hysteresis loop measurements can provide a reliable nanoscopic insight into the local potential variations due to defects, grain boundaries, etc. In other words, PFM hysteresis loops can in fact be very useful in elucidating microscopic details that cannot be determined from the integral macroscopic P-V loops alone. For example, a difference in the microscopic mechanisms related to the strongly imprinted ferroelectric capacitors and antiferroelectric capacitors, which exhibit similar macroscopic P-V hysteresis loops, has been previously provided by PFM [xv].

In PFM, parasitic effects related to the charge injection can be obtained by plotting the differential piezoresponse obtained by subtracting the raw piezoresponse (*Amplitude*cos(phase)*) in the bias-off mode from the corresponding signal in the bias-on mode (Figures R3(a,b,d,e)). Typically, this differential signal is linearly proportional to the applied voltage and exhibits a negative slope, but with additional contributions, such as charge injection, the differential loop acquires a non-linear behavior as shown in Figure R3(c) [xvi]. Such non-linear behavior was not observed in our TiN/La:HfO₂/TiN capacitors signifying the lack of extrinsic contributions to the measured signal (Figure R3(f)).

Figure R3: Identifying presence of charge injection from PFM. (a-c) PFM phase (top panel) and amplitude (bottom panel) loops in the bias on (a) and bias off (b) modes and differential piezoresponse loops (c) in IrO₂/La:HfO₂/IrO₂ capacitors. (d-f) PFM phase (top panel) and amplitude (bottom panel) loops in the bias on (d) and bias off (e) modes and differential piezoresponse loops (f) in TiN/La:HfO₂/TiN capacitors. The differential piezoresponse loops were obtained by subtracting the bias off piezoresponse signal from the bias on piezoresponse signal.

Furthermore, we would like to stress that PFM has been successfully used to elucidate the microscopic properties of a number of polycrystalline ferroelectric materials, such as PZT [xvii, xviii, xix, xx] and BiFeO₃ [xxi]. There is no reason to doubt the method. One just has to perform careful consideration of various potential parasitic contributions to the measured PFM signal to get correct interpretation of the acquired data. And this is exactly what has been done in this manuscript. Hence, we strongly disagree with the reviewer's assessment regarding the validity of interpretation of the obtained PFM results, which are indeed definitive contrary to the referee's opinion.

References

-
- ⁱ Buragohain, P., Richter, C., Schenk, T., Lu, H., Mikolajick, T., Schroeder, U., and Gruverman, A. Nanoscopic structure of domain structure dynamics in ferroelectric La:HfO₂ capacitors. *Applied Physics Letters*, 112, 222901 (2018).
- ⁱⁱ Lim, S. Y., Park, M. S., Kim, A., and Yang, S. M. Nonlinear domain wall velocity in ferroelectric Si-doped HfO₂ thin film capacitors. *Applied Physics Letters*, 118, 102902 (2021).
- ⁱⁱⁱ Stolichnov, I., Cavalieri, M., Colla, E., Schenk, T., Mittmann, T., Mikolajick, T., Schroeder, U., and Ionescu, A. M. Genuinely ferroelectric sub-1-Volt-switchable nanodomains in Hf_xZr_(1-x)O₂ ultrathin capacitors. *ACS Applied Material Interfaces* 10 (36), 30514-30521 (2018).
- ^{iv} Stolichnov, I., Cavalieri, M., Gastaldi, C., Hoffmann, M., Schroeder, U., Mikolajick, T., and Ionescu, A. M. Intrinsic or nucleation-driven switching: An insight from nanoscopic analysis of negative capacitance Hf_{1-x}Zr_xO₂-based structures. *Applied Physics Letters* 117, 172902 (2020).
- ^v Collins, L., Liu, Y., Ovchinnikova, O. S., and Proksch, R. Quantitative Electromechanical Atomic Force Microscopy, *ACS Nano* **13**, 7, 8055–8066 (2019).
- ^{vi} You, L., Zhang, Y., Zhou, S., Chaturvedi, A., Morris, S.A., Liu, F., Chang, L., Ichinose, D., Funakubo, H., Hu, W., Wu, T., Liu, Z., Dong, S., and Wang, J. Origin of giant negative piezoelectricity in layered van der Waals ferroelectric. *Science Advances* 2019, 5, :eaav3780.
- ^{vii} Starschich, S., Griesche, D., Schneller, T., Waser, R. and Böttger, U. Chemical solution deposition of ferroelectric yttrium-doped hafnium oxide films on platinum electrodes, *Applied Physics Letters* 104, 202903 (2014).
- ^{viii} Starschich, S., Schenk T., Schroeder, U. and Boettger, U. Ferroelectric and piezoelectric properties of Hf_{1-x}Zr_xO₂ and pure ZrO₂ films, *Applied Physics Letters* 110, 182905 (2017).
- ^{ix} Schenk, T., Godard, N., Mahjoub, A., Girod, S., Matavz, A., Bobnar, V., Defay, E. and Glinsek, S. Toward Thick Piezoelectric HfO₂-Based Films. *Phys. Status Solidi, RRL*, 14: 1900626 (2020).
- ^x Shimura, R., Mimura, T., Tateyama, A., Shimizu, T., Yamada, T., Tanaka, Y., Inoue, Y., and Funakubo, H. Preparation of 1 μm thick Y-doped HfO₂ ferroelectric films on

(111)Pt/TiO_x/SiO₂/(001)Si substrates by a sputtering method and their ferroelectric and piezoelectric properties. *Japanese Journal of Applied Physics* 60, 031009 (2021).

^{xi} Hong, S. Single frequency vertical piezoresponse force microscopy, *Journal of Applied Physics* 129, 051101 (2021).

^{xii} Kholkin, A., Akdogan, E. K., Safari, A. Chauvy P.-F, and Setter, N. Characterization of the effective electrostriction coefficients in ferroelectric thin films, *Journal of Applied Physics* 89, 8066-8073 (2001).

^{xiii} Furukawa, T., and Seo, N. Electrostriction as the origin of piezoelectricity in ferroelectric polymers. *Japanese Journal of Applied Physics*, 29, 675-680 (1990).

^{xiv} Katsouras, I., Asadi, K., Li, M., van Driel, T.B., Kjaer, K.S., Zhao, D., Lenz, T., Gu, Y., Blom, P.W.M., Damjanovic, D., Nielsen, M.M., and de Leeuw, D.M. The negative piezoelectric effect of the ferroelectric polymer poly(vinylidene fluoride)., *Nature Materials* 15, 78-84 (2016).

^{xv} Lu, H., Glinsek, S., Buragohain, P., Defay, E., Iñiguez, J., and Gruverman, A. Probing Antiferroelectric-Ferroelectric Phase Transitions in PbZrO₃ Capacitors by Piezoresponse Force Microscopy. *Advanced Functional Materials*, 30, 2003622 (2020).

^{xvi} Balke, N., Jesse, S., Li, Q., Maksymovich, P., Okatan, M.B., Streclov, E., Tselev, A., and Kalini, S. Current and surface charge modified hysteresis loops in ferroelectric thin films. *Journal of Applied Physics*, 118, 072013 (2015).

^{xvii} Gruverman, A., Rodriguez, B. J., Dehoff, C., Waldrep, J. D., Kingon, A. I., Nemanich, R. J., Cross, J. S. Direct studies of domain switching dynamics in thin film ferroelectric capacitors. *Applied Physics Letters* 87, 082902 (2005).

^{xviii} Nagarajan, V., Aggarwal, S., Gruverman, A., Ramesh, R., Waser, R. Nanoscale polarization relaxation in a polycrystalline ferroelectric thin film: Role of local environments. *Applied Physics Letters*, 86, 262910 (2005).

^{xix} Jo, J. Y., Han, H. S., Yoon, J. -G., Song, T. K., Kim, S. -H., and Noh, T. W. Domain Switching Kinetics in Disordered Ferroelectric Thin Films. *Physical Review Letters*, 99, 267602 (2007).

^{xx} Kholkin, A. L., Bdikin, I. K., Kiselev, D. A., Shvartsman, V. V., and Kim, S. -H. Nanoscale characterization of polycrystalline ferroelectric materials for piezoelectric applications. *Journal of Electroceramics*.19, 83-96 (2007).

^{xxi} Jin, Y., Lu, X., Zhang, J., Kan, Y., Bo, H., Huang, F., Xu, T., Du, Y., Xiao, S. and Zhu, J. Studying the Polarization Switching in Polycrystalline BiFeO₃ Films by 2D Piezoresponse Force Microscopy. *Scientific Reports*, 5, 12237 (2015).

REVIEWERS' COMMENTS

Reviewer #1 (Remarks to the Author):

This referee has carefully gone through the rebuttal, revised manuscript, and updated supplementary information. All concerns of this referee have been satisfactorily addressed and the publication of this work could be considered by the editors.

Reviewer #2 (Remarks to the Author):

The authors successfully answered my questions. Still, their theoretical explanations are not fully satisfactory, considering the complexity of this compound, I agree on the publication of the current manuscript in Nature Communications.

Reviewer #3 (Remarks to the Author):

The response letter was very carefully written discussing in details the main concerns of this referee. Therefore, the previous judgement is revised into a more positive assessment of the present work, suggesting the work to be published.